# Evolution, geographic spreading, and demographic distribution of Enterovirus D68

**Emma B. Hodcroft** [1,2,3]*, **Robert Dyrdak** [4,5], **Cristina Andrés** [6], **Adrian Egli** [7,8],
**Josiane Reist** [7,8], **Diego García Martínez de Artola** [9], **Julia Alcoba-Flórez** [9], **Hubert G. M. Niesters** [10], **Andrés Antón** [6], **Randy Poelman** [10], **Marijke Reynders** [11],
**Elke Wollants** [12], **Richard A. Neher** [1,3], **Jan Albert** [4,5]

**1** Biozentrum, University of Basel, Basel, Switzerland, **2** Institute of Social and Preventive Medicine, University of Bern, Bern, Switzerland, **3** Swiss Institute of Bioinformatics, Basel, Switzerland, **4** Department of Clinical Microbiology, Karolinska University Hospital, Stockholm, Sweden, **5** Department of Microbiology, Tumor and Cell Biology, Karolinska Institute, Stockholm, Sweden, **6** Respiratory Viruses Unit, Virology Section, Microbiology Department, Vall d'Hebron Hospital Universitari, Vall d'Hebron Institut de Recerca (VHIR), Vall d'Hebron Barcelona Hospital Campus, Barcelona, Spain, **7** Clinical Bacteriology and Mycology, University Hospital Basel, Basel, Switzerland, **8** Applied Microbiology Research, Department of Biomedicine, University of Basel, Basel, Switzerland, **9** Department of Clinical Microbiology, Hospital Universitario Nuestra Señora de Candelaria, Tenerife, Spain, **10** University of Groningen, University Medical Center Groningen, Department of Medical Microbiology, Division of Clinical Virology, Groningen, The Netherlands, **11** Unit of Molecular Microbiology, Medical Microbiology, Department of Laboratory Medicine, AZ Sint-Jan Brugge AV, Bruges, Belgium, **12** KU Leuven, Rega Institute, Department of Microbiology, Immunology and Transplantation, Laboratory of Clinical & Epidemiological Virology, Leuven, Belgium

* emma.hodcroft@ispm.unibe.ch

**Data Availability Statement:** All relevant data are within the manuscript and its Supporting information files or available on Github, VipR, and

## Abstract

Worldwide outbreaks of enterovirus D68 (EV-D68) in 2014 and 2016 have caused serious respiratory and neurological disease. We collected samples from several European countries during the 2018 outbreak and determined 53 near full-length genome ('whole genome') sequences. These sequences were combined with 718 whole genome and 1,987 VP1-gene publicly available sequences. In 2018, circulating strains clustered into multiple subgroups in the B3 and A2 subclades, with different phylogenetic origins. Clusters in subclade B3 emerged from strains circulating primarily in the US and Europe in 2016, though some had deeper roots linking to Asian strains, while clusters in A2 traced back to strains detected in East Asia in 2015-2016. In 2018, all sequences from the USA formed a distinct subgroup, containing only three non-US samples. Alongside the varied origins of seasonal strains, we found that diversification of these variants begins up to 18 months prior to the first diagnostic detection during a EV-D68 season. EV-D68 displays strong signs of continuous antigenic evolution and all 2018 A2 strains had novel patterns in the putative neutralizing epitopes in the BC- and DE-loops. The pattern in the BC-loop of the USA B3 subgroup had not been detected on that continent before. Patients with EV-D68 in subclade A2 were significantly older than patients with a B3 subclade virus. In contrast to other subclades, the age distribution of A2 is distinctly bimodal and was found primarily among children and in the elderly. We hypothesize that EV-D68's rapid evolution of surface proteins, extensive diversity, and high rate of geographic mixing could be explained by substantial reinfection of adults. Better understanding of evolution and

Genbank using the links and accession numbers within the manuscript and supporting files.

**Funding:** This study was funded by the University of Basel through core funding (EBH, RAN) and a grant from the Swedish Foundation for Research and Development in Medical Microbiology (grant '2019-03-18') (RD). The funders had no role in study design, data collection and analysis, decision to publish, or preparation of the manuscript.

**Competing interests:** The authors have declared that no competing interests exist.

immunity across diverse viral pathogens, including EV-D68 and SARS-CoV-2, is critical to pandemic preparedness in the future.

## Author summary

Enterovirus D68 (EV-D68) has caused punctuated, global outbreaks of respiratory illness and neurological disease, including being implicated as the cause of acute flaccid myelitis (AFM). Serology studies and surveillance data suggests almost everyone is infected during early childhood. The majority of sequences collected are from young children, while adults retain high antibody titers against strains that circulated when they were young. However, little is known about how outbreaks are connected and how the virus evolves and spreads around the globe. Despite EV-D68's apparent reliance on young, naive hosts, EV-D68 antibody binding sites are reportedly evolving under antigenic pressure, and EV-D68 seems to spread rapidly during outbreaks. In this multi-center European collaboration, we confirm that subclade specific age differences are present in those infected. Further, we were able to quantify between- and within-country migration and the 'hidden' diversification that indicates unsampled circulation between outbreaks. We conclude that the evolution of EV-D68 may be driven by substantial re-infection of adults, explaining the rapid geographic mixing and continuous antigenic evolution. The presence of largely unsampled circulation prior to outbreaks suggests there are gaps in current surveillance practices which could be addressed by expanding genetic surveillance.

## Introduction

Enterovirus D68 (EV-D68) has caused worldwide outbreaks of serious respiratory and neurological disease in 2014 [1] and thereafter. In particular, EV-D68 infection has been associated with acute flaccid myelitis (AFM) [2].

EV-D68 was first isolated in 1962, [3] but rarely reported until the recent outbreaks [1, 4–6]. Yet, an almost ubiquitous presence of specific neutralizing antibodies indicates that infection with the virus has been very common before the recent outbreaks [7]. EV-D68 is an unenveloped, single-stranded RNA virus that primarily transmits through a respiratory route, though it can sometimes be detected in gastrointestinal samples. The common clinical manifestation of infection is an acute respiratory illness, which ranges from mild in most cases to severe enough for the need for supportive care in an intensive care unit. Less common are gastrointestinal symptoms, and rarely, the severe paralysis symptoms of AFM [1]. In many countries, particularly in Europe and North America, EV-D68 has exhibited a biennial pattern, with peaks in the late summers and autumns of even-numbered years (2014, 2016, 2018) [5, 8–13]. Other countries have reported odd-year outbreaks, such as Thailand in 2009–2011 [14], Australia in 2011 and 2013 (peaking in Southern Hemisphere winter to spring) [15], and Japan in 2015 [16] (see regional sequencing patterns in S7 Fig). Nonetheless, a modelling of estimated incidence in US showed that the biennial pattern of EV-D68 might be a transient feature [17]. This would be similar to other non-polio enteroviruses, where patterns are driven by the time it takes for a sufficiently large susceptible cohort to emerge [18], which may vary in different settings and/or over time. In 2018, circulation was reported from Europe in Wales [19], Italy [20], France [21], and in the US [9, 22].

Phylogenetic analysis of EV-D68 sequences has revealed extensive diversity [23]. This diversity is grouped into the major clades A, B, and C, with a most recent common ancestor of all clades (MRCA) in the mid-1990s. Clades A and B are further divided into subclades A1, A2, B1, B2, and B3. Most previous phylogenetic analyses have focused on the capsid protein VP1 sequence [23, 24]. VP1 is one of the more variable proteins and contains important receptor binding sites and putative neutralizing epitopes, such as the hypervariable BC- and DE-loops. Different subclades differ at several positions in these loops, and previous studies show that they have an elevated rate of amino acid substitutions [25] and appear to be under positive selection [14, 24–27].

The great majority of reported EV-D68 cases are pediatric [1], and sero-positivity to EV-D68 increases rapidly during childhood, reaching ubiquity in adults [28–32]. Relatively little is known about immunity to EV-D68, but infection appears to elicit long-lasting strain-specific immunity, as evidenced by the high prevalence among adults of neutralising antibodies to the prototype Fermon strain from 1962 [29–31, 33]. While some studies indicate strain-specific differences in neutralizing titers in different age-groups, the degree to which EV-D68 evolution is driven by immune escape and whether incidence patterns are determined by pre-existing immunity is unclear.

Here, in the first study to sample and sequence whole-genome EV-D68 from across several European countries, we report sequences from late-summer and autumn 2018 and combine them with publicly available data to comprehensively investigate the global phylodynamics and phylogeography of EV-D68. Continuously updated data and phylogenies are available on the Nextstrain platform at nextstrain.org/enterovirus, which provides a comprehensive phylogenetic analysis pipeline, a visualization tool and a public web-interface [34].

## Materials & methods

### Ethics statement

The study aimed to characterize the genetic diversity and geographic distribution of EV-D68 variants in Europe during the 2018 season and to compare these variants with EV-D68 variants collected earlier across the world. Respiratory samples (n = 55) that had tested positive for EV-D68 were obtained from six clinical virology laboratories (Stockholm, Sweden; Groningen, the Netherlands; Leuven, Belgium; Barcelona, Spain; Basel, Switzerland; and Tenerife, Canary Islands, Spain, see Table 1).

Ethical approvals were obtained as stipulated by national and local regulations on research ethics. For details see see section A in S1 Text. In short, no informed consent was needed for samples from Sweden, the Netherlands, Switzerland and Barcelona, Spain because the samples

**Table 1. 2018 Samples Sequenced.**

| Lab | Country | Number of Samples |
|---|---|---|
| Hubert Niesters et al. | Netherlands | 10 |
| Elke Wollants et al. | Belgium | 10 |
| Adrian Egli et al. | Switzerland | 6 |
| Andrés Antón et al. | Spain | 14 |
| Diego García Martínez de Artola | Spain | 8 |
| Jan Albert et al. | Sweden | 5 |

This refers to samples included in analyses and may not include samples received that did not achieve sufficient coverage to be included.

were anonymised. In Belgium no ethical approval was needed because the laboratory was a national reference laboratory on enteroviruses. In Tenerife, Spain an approval was obtained from the Local Research Ethics Committee (code CHUNSC 2019 02).

## Characteristics of patients and samples

Samples were collected from 29 Aug to 28 Nov 2018 (mean 4 Oct, median 1 Oct, interquartile range 22 Sept to 15 Oct, see S1 Fig). Thirty-two of the 53 successfully sequenced patients were male (60.4%). The median age was 3.0 years (mean 20.8 years, range 1 month to 94 years), with a majority of children being five years or younger (34 of 36 children), and a majority of adults being older than 50 years (13 of 17 adults). 39 of 43 patients with clinical information presented with respiratory illness as the main symptom. As EV-D68 is primarily a mild disease, most cases in a population are not expected to seek care or be sampled. Of 38 patients with admission status information, 32 were hospitalized (5 admitted to an intensive care unit (ICU)) and 6 were out-patient. There were no records of AFM among the study patients. Detailed information about the samples is available in S1 Table. For a more detailed description of EV-D68 PCR testing, sample selection and handling, see section A in S1 Text.

## EV-D68 sequencing and analyses

Near full-length genomes (NFLG) were obtained by sequencing four overlapping fragments on the Illumina next-generation sequencing (NGS) platform as described previously [25] (53 of 55 samples were successfully sequenced and included in the further analyses). Sequences were aligned with and annotated according to the 1962 Fermon strain (GenBank accession AY426531). See section B of S1 Text for details.

We combined the new 53 EV-D68 genomes with sequences available from GenBank and consolidated available metadata. This included 718 genome sequences with length $\geq 6,000bp$ and 1,987 sequences covering $\geq 700bp$ of the VP1 region of the genome (see section C of S1 Text). We analyzed the combined sequence data set with the Nextstrain pipeline [34]. The detailed analysis workflow is described in section E of S1 Text. A breakdown of the number of VP1 and whole genome sequences per year per region can be found in S7 and S8 Tables.

## Results

### Interactive near real-time phylogenetic analysis with Nextstrain

To enable global genomic surveillance of EV-D68, we implemented an automated phylogenetic analysis pipeline using Nextstrain which generates an interactive visualization integrating a phylogeny with sample metadata such as geographic location or host age. This analysis is available at nextstrain.org/enterovirus and was updated whenever new data became available and we intend to keep it up-to-date going forward. This rolling analysis revealed a dynamic picture of diverse clades of EV-D68 circulating globally.

Fig 1 shows a time-scaled phylogenetic tree of all available VP1 sequences ($\geq 700bp$) collected since 1990. Since 2014, the global circulation of EV-D68 has been dominated by viruses from the B1 and B3 subclades, with A1 and A2 accounting for about 5 to 30% of viruses sampled in China, Europe, and Africa (see bottom of Fig 1). Both our 53 new sequences and other available EV-D68 variants from 2018 grouped into the B3 and A2 subclades.

### Multiple independent origins of the 2018 season

As shown in Fig 1, VP1 sequences collected in 2018 fell into multiple distinct subgroups with MRCAs in 2016 or more recently. Six of these subgroups and one singleton fell into subclade

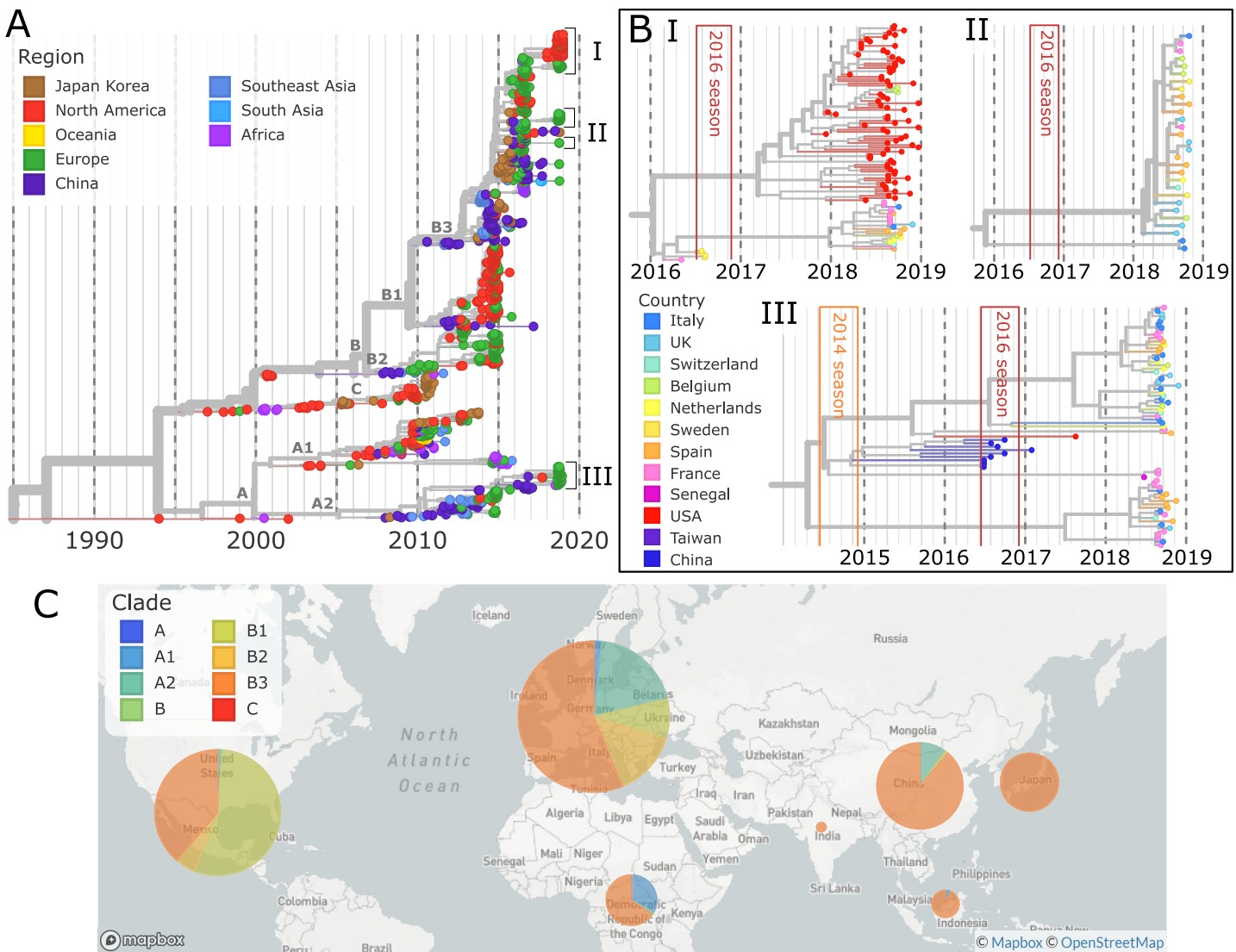

**Fig 1. Time-scaled phylogeny of Enterovirus D68. A)** A time-scaled phylogeny of VP1 segments, colored by region, is shown top-left. **B)** Key clusters (I, II, III), the largest from the 2018 season, are highlighted, colored by country. To display these clusters at a high resolution, the whole genome phylogenies are used. The 2014 and 2016 EV-D68 seasons are shown in orange and red boxes. **C)** A map shows the distribution of subclades by region from 2014–2018. Map from Mapbox and OpenStreetMap.

B3, while another four subgroups were part of subclade A2. The largest B3 subgroup (zoomed at the top of Fig 1) consisted of 80 samples from the USA and 3 Belgian samples, while most remaining subgroups were smaller and dominated by European samples.

The B3 subgroups from 2018 mostly originate from within the EV-D68 sequence diversity that circulated during the 2016 outbreak. This is expected as the B3 subclade was relatively well-sampled in Europe, North America, China, and Japan during 2015–2016, with more VP1 sequences in this period than any other clade in any period. The 2018 subgroups did not necessarily have ancestors in 2016 from the same geographic regions. Instead, some subgroups sampled in Europe in 2018 had their closest relatives in 2016 in Asia (see Fig 1A and the online phylogeny for more detail). For example, all 2018 A2 subclade samples traced back to share

common ancestors with strains circulating in China in 2015–2016 (Fig 1B, III). These observations suggest relatively rapid global mixing of EV-D68.

## Undetected EV-D68 diversification

Our phylogenetic analysis suggests that many of the 2018 subgroups started to diversify as early as mid-2017 but were not sampled until about one year later (Fig 1A and 1B). Both of the large, predominantly European 2018 subgroups have estimated MRCAs in the middle to end of 2017, with extensive diversification until the first samples were collected in August of 2018. The large American 2018 subgroup began diversifying in early 2017. Though the ancestors of these subgroups must have been circulating, there are no corresponding samples. This pattern is not unique to the B3 subclade or to 2018. Similar patterns of 'hidden' diversification can be seen in the two large 2018 clusters in the A2 subclade, and in the 2014 and 2016 outbreaks, with the ancestors of samples taken during these outbreaks beginning to diversify up to a year beforehand.

We quantified this diversification over time in Fig 2B, where the number of lineages which lead to the samples taken during each season is plotted over time [35] (see section F of S1 Text). Despite a variable number of lineages present at the end of each outbreak (363 for 2014, 119 for 2016, and 122 for 2018), each season traces back to only 10–18 lineages two years prior, reflecting a marked diversification which began about 18 months before the first sequenced samples of each season were collected. Four years prior to the outbreak year the number of lineages dropped further, to 2–7, representing deep splits in the tree.

When the change in number of lineages is plotted alongside sampling times (Fig 2C), peak diversification (black) appears before peak sampling periods (purple) and stretches back substantially to before the outbreak. We expect that the diversity of the samples, and the speed at which this diversity was generated, is proportional to the width and height of the diversification plots. Both plots illustrate that the majority of diversification occurs prior to when samples are taken (i.e. 'hidden' diversification), and slows down during the outbreak. This pattern is

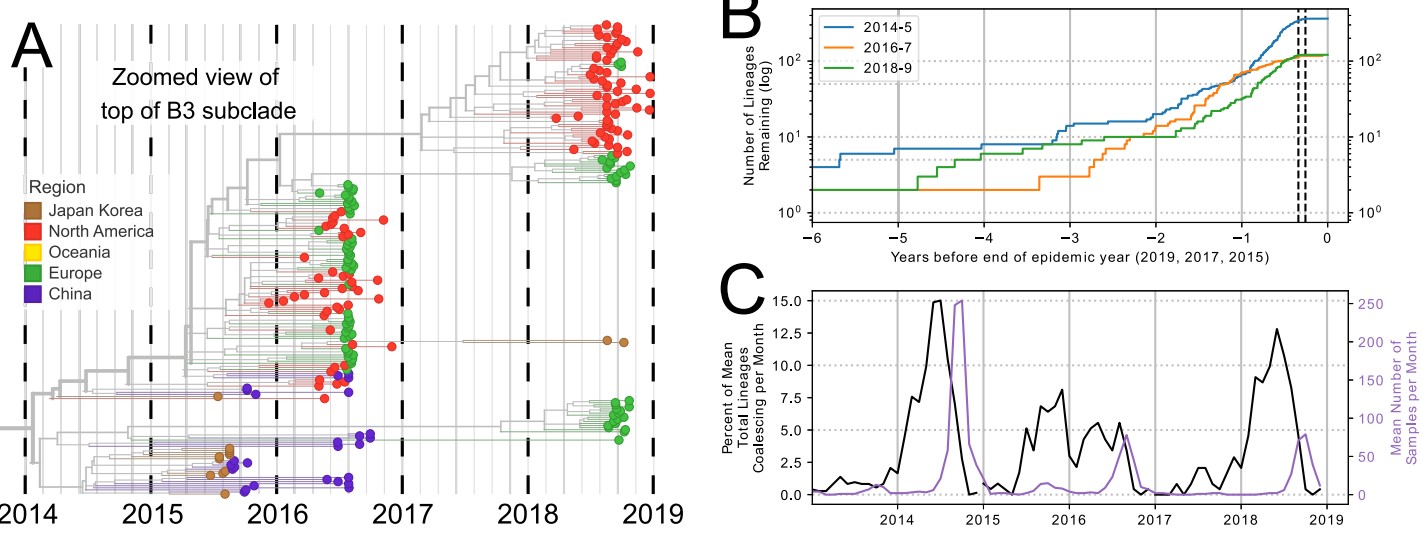

**Fig 2. Persistence and diversification of EV-D68 since 2014.** Multiple EV-D68 lineages persist from one biennial outbreak to another. (A) Zoomed-in view of the whole genome EV-D68 tree (for better time-resolution), showing 2018 subgroups in the B3 subclade. (B) Lineages which are ancestral to all of the samples in the 2014, 2016, and 2018 seasons diversify over time. The dotted black lines show the IQR of all samples taken during 2014, 2016, and 2018. (C) The change in number of lineages (as % of total lineages) per month for each season (left y-axis) and the mean number of samples per month (right y-axis).

consistent with the expectation that lineages split in expanding populations and little coalescence is observed when populations are large [36]. In the 2014 and 2018 seasons, the majority of diversification immediately preceded the sampling period, evident in the steep slopes in the first half of the outbreak year in Fig 2B, and in the sharp peaks during the outbreak year in Fig 2C. In the 2016 outbreak, however, the diversification appears to have been slower, shown by the shallower but more stable slope in Fig 2B and the broad peak of lineage change in Fig 2C. In 2014 and 2018, diversification began approximately a year before the outbreak period, whereas in 2016, it began almost 18 months before. These patterns are recapitulated in the inferred pattern of the per-lineage coalescence rate through time ('skyline,' see S3 Fig). The well-sampled seasons in 2014, 2016, and 2018 show a common pattern: diversity within an outbreak traces back to 10–18 lineages 2 years prior and 2–7 lineages 4 years before the outbreak.

Overall, our results suggest that EV-D68 diversity is maintained through multiple outbreak years by significant unreported year-round circulation.

## Migration of EV-D68 between countries and continents

The relationship between 2016 and 2018 viruses suggested that EV-D68 transmits and migrates rapidly enough that viruses sampled in one continent in 2018 often have ancestors detected in 2016 on different continents. To quantify the geographical mixing we investigated the viral migrations between countries and continents during the comparatively well-sampled outbreaks of 2014, 2016 and 2018 (see section F of S1 Text).

Of the 10 outbreak clusters in 2018, all but two were predominantly European. Within these clusters, strains from different European countries were thoroughly intermixed, with little sign of within-country clustering (see insets in Fig 1). Consistent with this qualitative observation, the maximum likelihood estimate of the overall migration rate between European countries in 2018 was approximately 2/year. Repeating the same analysis on the 2014 and 2016 outbreaks yielded about 2- to 3-fold lower estimates, possibly due to less representative sampling in these years. Discrete trait analyses make a number of assumptions and the migration rate estimates should be viewed as an empirical summary of transitions observed on the tree rather than unbiased quantitative estimates.

At the level of broader geographic regions, mixing was less rapid. This is qualitatively evident from the clustering of regions in the phylogeny in Fig 1. The maximum likelihood migration rate between Europe, North America, and China estimated using a discrete state model fitted to years 2014–2018 was found to be 0.24/year. This means that any given lineage has a chance of about 1 in 4–5 to switch regions in one year. After the two year inter-outbreak interval, about half the viral lineages would be expected to have spread to other geographic regions. Notably, though samples from the 2014 and 2016 outbreaks showed considerable mixing between European and North American sequences [25], all 77 samples from the US in 2018 formed one distinct subgroup, containing only three samples from elsewhere (Belgium). Since the subgroup dominating the US in 2018 harbors substantial diversity and since both the US and Europe have been relatively well-sampled (S7 Table), these divergent patterns are difficult to explain with sampling biases alone.

## Subclade A2 is over-represented in the elderly

We noted that many samples in the A2 subclade were obtained from adults and the elderly, in contrast to other subclades. Similar observations have been made in some previous studies [21, 27, 37–39]. Therefore, we investigated the age distribution in different subclades using a comprehensive data set of 743 samples with exact (non-range) age data and unambiguous subclade

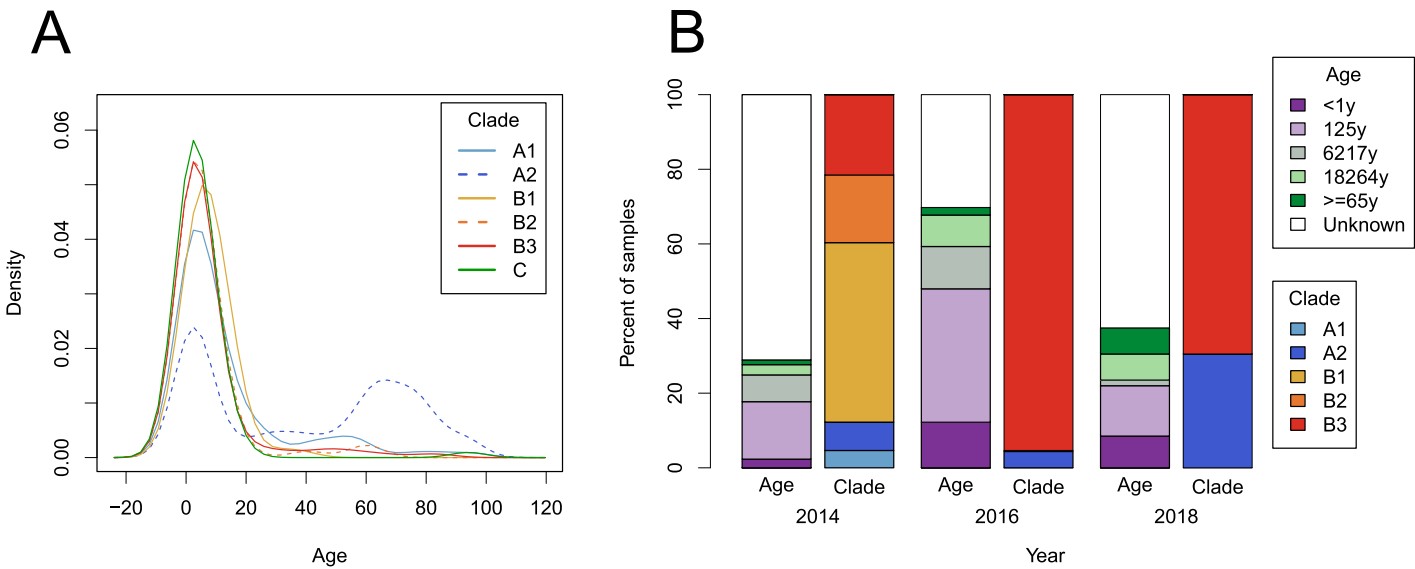

**Fig 3. Age distribution of EV-D68 samples.** (A) The cumulative distribution over age separated by subclade; A2 being significantly more often detected in older persons. (B) The age of patients and subclade of their sample, for samples taken during 2014, 2016, and 2018 for which 'age range 1' (see section D of S1 Text) was available.

designation (see section D of S1 Text). Almost all sequences assigned to B or C clades were sampled from children, while subclade A2 viruses showed a bimodal age distribution (Fig 3A), with 40% of A2 samples from children and 39% from the elderly (>60 years old). Subclade A1 was sampled predominantly in children with a minority of sequences sampled from individuals between 20 and 60 years of age. The distribution of age categories in each subclade is shown in S4 Fig.

These associations are confirmed by statistical analysis. Both the A1 and A2 subclades were associated with significantly older ages compared to B3 subclade (linear model, p = 0.001 and p<0.001, respectively), whereas individuals infected with B2, B1, and C subclades did not have significantly different ages (Table 2). When sample year or region were added individually as co-factors, to control for possible differences in sampling patterns, the association remained significant (p<0.001). While these p-values might by inflated due to regional differences in sampling different age groups, the difference in age distribution between the A2 and B subclades was robust to removal of countries by bootstrap analysis (S2 Fig). In addition to the age differences between the A and B subclades, the age distributions of the A1 and A2 subclades also differed significantly ($p < 0.001$, Kolmogorov-Smirnov test).

**Table 2. Age and Clade/Subclade.** Linear model of subclade on age, showing the ages of patients in subclades A2 and A1 are significantly different from those in subclade B3. Subclades B2, B1, and C do not differ significantly from B3. Number of samples refers to the number of samples within a subclade for which exact age data was available.

| (Sub)Clade | Number of Samples | Mean Age | Differs from Intercept |
|---|---|---|---|
| B3 (Intercept) | 371 | 7.33 | |
| B2 | 59 | 6.54 | NS ($p = 0.74$) |
| B1 | 73 | 6.64 | NS ($p = 0.75$) |
| A2 | 79 | 38.13 | *** $p < 2 \times 10^{-16}$ |
| A1 | 88 | 13.66 | ** $p = 0.002$ |
| C | 75 | 4.9 | NS ($p = 0.25$) |

The higher prevalence of the A2 subclade in 2018 as compared to previous outbreaks results in a significant skew towards older age groups in 2018 compared to 2014 and 2016 (linear model, p = 0.005 and p = 0.001, respectively) (Fig 3B).

## Antigenic evolution of EV-D68

Previous molecular analyses of EV-D68 showed the BC- and DE-loops in the VP1 gene have elevated rates of amino acid substitutions [25] and indications of positive selection [14, 24, 26, 27]. We examined the sequences of these putative neutralizing antibody targets [40] over time, comparing the sequences of the loops at the root of the tree in 1990, at the base of the A and B clades, and from several sequences sampled in 2018 (Fig 4B). Both loops have changed their sequence multiple times in each of these comparisons.

Still, even outside of these loops the surface of the virus has changed rapidly, suggesting further antigenic evolution. Fig 4A shows the 5-fold symmetric arrangement of the crystal structure of the virus capsid protomer consisting of VP1, VP2, VP3, and VP4 [41]. Much of the virion surface is variable (panel A, subunit 5), and almost all variable residues in VP1 and VP2 are in surface exposed patches of the protein sequence (panel C). A particularly dense cluster of variable positions is formed by the C-terminus of VP1 and central region of VP2. The C-terminus of VP1 was identified by Mishra *et al.* [42] as a potential EV-D68 specific epitope. In VP1 to VP4, there are 6 times as many mutations per site at residues on the outer surface compared to those buried or on the interior of the capsid (OR = 6, $p < 10^{-16}$). This remains significant if the genes are inspected individually, with the weakest signal in VP3 (OR = 2.7, $p = 0.0032$). The reduced variation in the 'canyon' residues (those less exposed to the surface, such as codons VP1:146–155 and VP3:132–141) matches observations in other picornaviruses [43].

When synonymous vs non-synonymous changes within VP1 are plotted (S9(A) Fig), the number of synonymous changes does not vary much, as expected, with most sites having between 5–10 mutations. The number of non-synonymous changes is quite different: most sites have very few changes, but some change more than 15 times. Codons within the BC- and DE-loop are among the sites with the most non-synonymous changes (S9(B) Fig).

The epitope turnover dynamics are more readily apparent when integrated with the phylogenetic tree, and trees colored by the epitope patterns found in the BC- and DE-loops reveal signs of rapid and putatively antigenic evolution in EV-D68 (Fig 4D and 4E). The BC-loop in the A2 subclade, for example, has changed 3 times since 2010 (a zoomed version of the D and E panels labelled with the number of changes can be seen in S8 Fig). All 2018 A2 samples have novel epitope patterns in the BC-loop (the two small grey 'other' clusters also have distinct novel patterns). The predominantly American B3 2018 subgroup also has novel BC-loop patterns, mostly pattern 'DTTQTF'. A total of 144 of 197 samples from 2018 have BC-loop sequences different from those observed in 2016 or earlier. The DE loop has undergone less rapid change in the recent past, but all major clades differ in their DE-loop sequence. The predominant epitope pattern in the mostly-American B3 subgroup ('NGSNNNTTYV') was present in 25 European sequences in 2016, but does not appear to have previously been seen in the USA. Notably, the B1 and B3 variants dominating the large outbreaks in 2014 and 2016 show little variation at the BC- or DE-loops. S5 Fig plots the mean number of mutations in the BC- and DE-loops acquired over time and shows that different subclades have acquired mutations at different rates. The patterns of molecular evolution revealed here are compatible with the notion of rapid antigenic evolution with rapid turnover of epitopes and frequent parallel changes in different clades, as observed for seasonal influenza viruses [44].

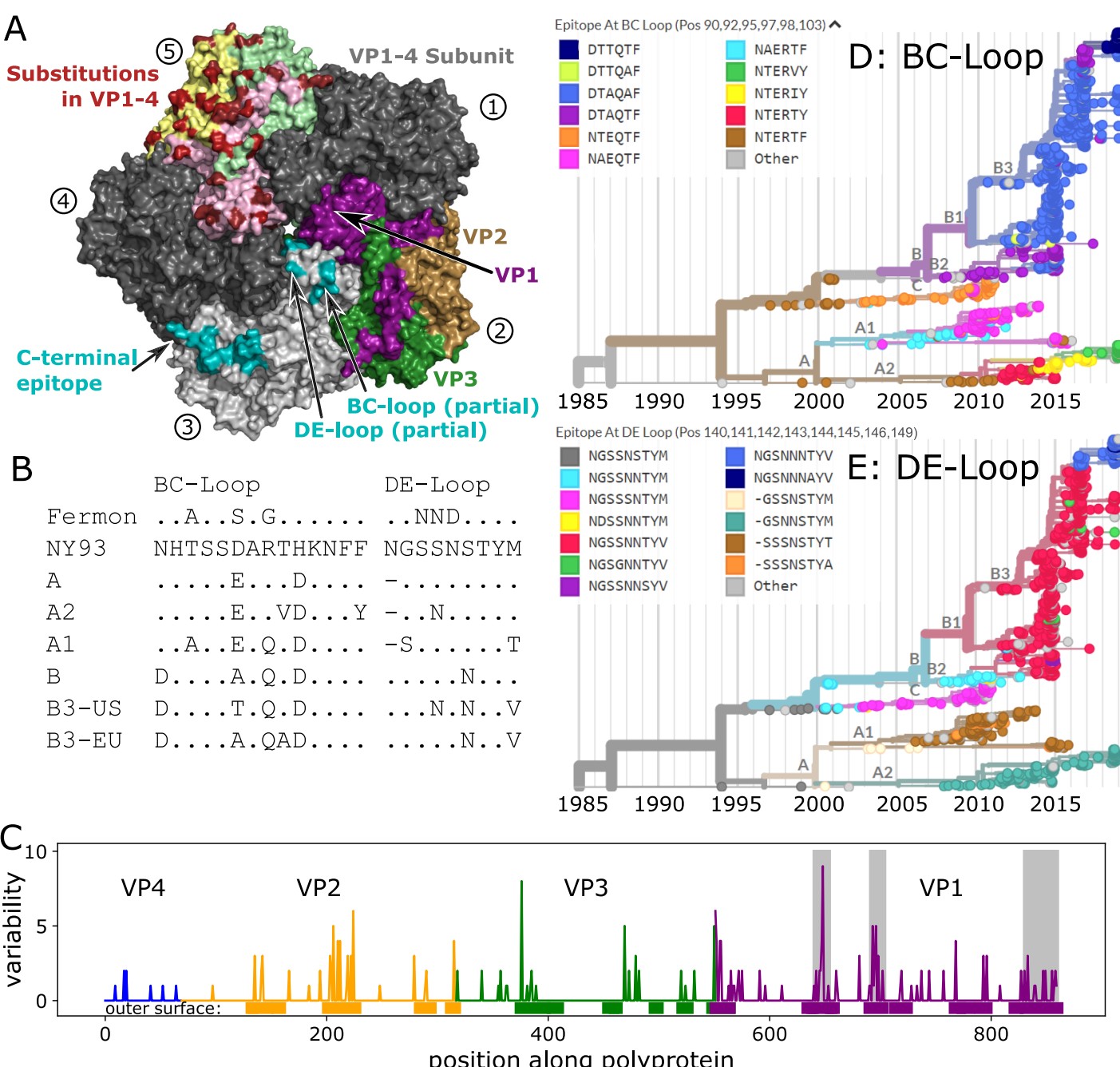

**Fig 4. Molecular evolution of the EV-D68 capsid. Panel A)** Rendering of 5-fold symmetric arrangement of the crystal structure of the capsid protomer consisting of VP1, VP2, VP3 and VP4 [41], using different copies of the protomer to highlight different aspects of the capsid organization, evolution, and immunogenicity. The five subunits are labelled 1–5 (circled numbers). Subunits 1 and 4 are present in dark grey to complete the structure. Subunit 2 shows the surface exposed proteins in purple (VP1), sand (VP2), and green (VP3), subunit 3 displays the different putative epitopes [42] (note that the most variable parts of the BC- and DE-loops are not shown, as their structure is not resolved), and subunit 5 highlights variable positions in red, with the different proteins forming the surface indicated by pale colors. **Panel B)** The hypervariable BC- and DE-loops (partly missing from the structure in Panel A) in clusters and subclusters accumulated multiple changes since the root of the tree (the inferred root sequence patterns match those of the NY93 strain shown here). (BC-loop is AA positions 90–103; DE-loop is AA positions 140–148). Patterns for the C-terminus and a region of VP2 are shown in S2 Table. **Panel C)** Variable positions often fall into surface exposed parts of the protein (Fisher exact test, OR = 6, $p < 10^{-16}$). Genes are highlighted by blue (VP4), orange (VP2), green (VP3), and purple (VP1). Grey boxes show, from left to right, the BC- and DE-loops, and C-terminus regions. **Panels D&E)** Phylogenetic VP1 trees are colored by the most common epitope patterns of the 6 and 8 most variable amino-acid positions in the BC- and DE-loops, respectively. Particularly note-worthy is the rapid turnover of BC-loop variants in the recent evolution of the A2 subclade. Patterns for the C-terminus region are shown in S6 Fig. See S8 Fig for a zoomed version of panels D & E with cumulative AA changes.

## Discussion

Through a European multi-center collaboration and use of publicly available sequences and metadata in Genbank, we have shown that the circulation of EV-D68 in 2018 was not a single outbreak but consisted of smaller clusters of diverse viruses from the A2 and B3 subclades. We demonstrated a robust association of A2 viruses with infections in the elderly and provide evidence for rapid antigenic evolution of the EV-D68 surface proteins.

### Recent EV-D68 circulation

We compared genetic diversity of viruses circulating in 2018 with those from the major US/European outbreaks in 2014 and 2016. In each of these seasons, sampled diversity traced back to 10–18 lineages at the time of the previous season two years earlier. Four to five years prior to a season, the number of ancestral lineages is reduced to 2–7. This suggests that EV-D68 has undergone waves of diversification about 1 year prior to the recognised outbreaks and that coalescence occurred on a 2–4 year time scale. Similar patterns can be observed for seasons prior to 2014, but the scarce sequence availability makes quantification of these patterns less reliable.

### Geographic mixing

All major subgroups circulating in Europe in 2018 were sampled in multiple countries, despite the fact that the inferred $T_{MRCA}$ of most subgroups was only about one year prior to sampling. Consistent with this, we inferred rapid migration of lineages between European countries at rates of about 2/year. While the great majority of sequences in this dataset were obtained from respiratory samples, Majumdar *et al.* [45] reported 27 EV-D68 sequences from wastewater samples from the UK (accession numbers MN018235-MN018261) obtained through Sanger sequencing. These sequences were well mixed with the European respiratory samples, indicating that the viruses detected in sampling in medical settings fairly accurately reflect the circulating strains.

At the level of continents or major geographic regions, migration was slower and accurate quantification difficult. Discrete trait analysis suggested migration rates between continents to be 5–10 fold lower than between European countries. While we observed rapid mixing between sequences sampled in the US and Sweden during the 2016 outbreak [25], sequences sampled in the US in 2018 formed a largely US-specific cluster.

Despite the (currently) monophyletic origin of 2018 EV-D68 in the US, EV-D68 strains are remarkably well-mixed given that the majority of cases were in young children.

### Age distributions

We undertook a thorough literature review and reached out to many authors to obtain information on the age of the sampled individuals. In total, we were able to assign ages to 743 sampled patients which allowed us to investigate age distributions with much higher accuracy than previous studies. We found that subclade A2 was sampled significantly more often in the elderly than other clades and subclades, which is consistent with results from previous smaller studies [21, 27, 37, 38]. More specifically, we found that A2 samples had a pronounced bimodal age distribution (Fig 3A) with about 40% of the samples from children and 40% from the elderly above 60 years. Notably, the A1 subclade was also moderately over-represented in adults, but not specifically in the elderly.

The fact that almost all samples from the elderly population fell into the A2 subclade could have several reasons. (i) Individuals in this age group could be more prone to infection by A2

subclade viruses, possibly due to lower subclade-specific immunity. This possibility is supported by Harrison *et al.* [30] who showed lower prevalence of neutralizing antibodies among adults to the A2 subclade than to B1, B2 and Fermon. (ii) Alternatively, A2 viruses are more virulent/symptomatic in elderly (irrespective of pre-existing immunity), but there are no data directly supporting this possibility.

## Evidence for continuous antigenic evolution

Given that most EV-D68 cases are pediatric [1] and that almost everyone has high EV-D68 antibody titers by the age of ten [28–31], one might expect that selection for antigenic change is of minor importance for the molecular evolution of EV-D68. However, we found rapid evolution in putative neutralizing epitopes, including the previously studied BC- and DE-loops [24, 25]. In particular, the C-terminus of VP1 and a surface exposed patch in VP2 are accumulating amino acid substitutions at a rate higher than the rest of the genome, but similar to the BC- and DE-loop epitopes, which are known to evolve more quickly, with substantial parallelism between clades. The C-terminus of VP1 has been described as an EV-D68 specific immunoreactive epitope [42]. In the 3D-structure, the C-terminus of VP1 and VP2 forms a contiguous ridge, which is a plausible target for antibody binding that displays substantial differences between the A2 and B3 subclades.

Antigenic evolution is further supported by serology. Harrison *et al.* [30] used a panel of sera collected in 2012 (prior to the large clade B outbreak in 2014) and measured neutralizing antibody titers against the prototypic Fermon strain from 1962 and against viruses from subclades B1, B2, and A2 sampled in 2014. Individuals older than 50 (i.e. born before 1962) have uniformly high titers against the Fermon strain (median $log_2$ titer 10.5), while children below the age of 15 have low titers ($log_2$ titer 6). In contrast, titer distributions against representatives from subclades B1 and B2 were similar for individuals above the age of 5 (median between 7.8 and 9.5). Young children below the age 5, however, had markedly lower titers (median 5.8 (B1) and 3.2 (B2)) suggesting that a substantial fraction of those under 5 years old are still naive. In pregnant women tested from 1983 to 2002, all showed continuously high EV-D68 seroprevalence to Fermon, but on a population level titers declined over the study period [33]. Taken together, this implies that individuals can retain high titers against a strain they were exposed to decades earlier (as shown by the titers of those over the age of 50 against Fermon, which has not circulated for many years), but that infection with recent EV-D68 strains does not elicit antibodies against Fermon—that is, the virus has evolved antigenically. Consistent with this, Imamura *et al.* [26] showed that antisera raised in guinea pigs against viruses from subclades A2, B2, and C, did not neutralize Fermon.

## Antigenic evolution could be driven by substantial re-infection of adults

While EV-D68 has historically been primarily diagnosed in children, unrecognised (re-)infection of adults could explain two striking features of EV-D68 diversity and evolution: (i) consistent patterns of molecular evolution suggesting rapid immune-driven antigenic evolution (as observed in influenza A/H3N2 [44]) and (ii) the rapid geographic mixing of different EV-D68 variants. The speed of antigenic evolution of influenza viruses has been negatively associated with the fraction of infection in children [46], and adults travel much more than children, allowing more migration opportunities.

For pre-existing immunity to be a driving force of viral evolution, reinfection of older, previously exposed individuals has to be sufficiently common. In adults with high antibody titers to EV-D68, post-outbreak titers against the circulating strain increase, implying both that existing immunity is not fully protective and that it can be boosted by subsequent re-exposure

[29]. Adults are less likely to attend medical care than children and infections in adults may be milder, further reducing the likelihood that they are sampled and explaining why diagnoses and sequences come predominantly from children. Less-symptomatic infections in adults may be due to both physiology (e.g. children have smaller respiratory tracts) and partially protective immunity from previous EV-D68 infection.

Increases in EV-D68 infection in particular groups of adults, as we see in the A2 subclade and individuals over 60, may be due to 'original antigenic sin,' whereby immune response is primarily based on the first pathogen encountered [47]. If new EV-D68 strains evolve to be sufficiently distant from the initial EV-D68 strain that infected a particular age cohort, these individuals could be particularly vulnerable to re-infection and/or more severe infections.

While existing serological data are consistent with continuous antigenic evolution and frequent infection of adults, currently available serological studies typically include only a small number of EV-D68 strains (antigens), making systematic comparisons of preexisting immunity to different EV-D68 clades difficult. Further serological work, which could be guided by considering what lineages age groups likely encountered early in life and the phylogenetic relationship and genetic distance to later lineages, will be critical in exploring this hypothesis for the role of original antigenic sin in EV-D68 reinfection.

## Conclusion

Platforms like nextstrain.org allow the near real-time analysis and dissemination of sequence data integrated with relevant metadata, which we hope will be a valuable resource to the community. In this comprehensive analysis of new and existing EV-D68 data, we reveal a very dynamic picture of EV-D68 circulation, characterized by rapid mixing at the level of countries within Europe, rapid evolution of surface proteins, and divergent and at times bimodal age distributions of different subclades. Combined with published serological data, our findings suggest a substantial contribution of adults in the global dispersion and continuous antigenic evolution of EV-D68.

The current SARS-CoV-2 pandemic has highlighted the importance of building our knowledge and understanding of virus transmission, evolution, and immunity. Through studying the interplay of human immunity and evolution of endemic viruses such as EV-D68, we can expand our knowledge of viruses to both better understand those circulating today and those we may face in the future. Our work on EV-D68 highlights that in endemic pathogens, antigenic escape may mean that early exposure is not sufficient to provide perfect protection for life, and critically, may allow previously infected adults to serve as transmission reservoirs and drivers of antigenic evolution. It is impossible to predict which pathogen may lead to the next viral pandemic, but by continuing to uncover the transmission dynamics, evolution, and long-lasting immunity of epidemic and endemic viruses we can have a better chance of more quickly combating new pandemic pathogens.

## Supporting information

**S1 Text. Supplementary methods.** Detailed descriptions of the patient populations and sampling methods, datasets, sequencing, bioinformatics, and phylogenetic analysis, and the analyses of age, diversity, and geographic spread.
(PDF)

**S1 Fig. Histogram of the sampling dates, by country, for the 53 samples generated in this study that were included in the final analyses.**
(PDF)

**S2 Fig. For the VP1 dataset, samples were sampled with replacement by country 100 times.** For each bootstrapped dataset, the same linear regression shown in Table 2 was performed, and the age distributions for the A1, A2, and B3 clades was plotted. The ages of the A2 clade was significantly different from the B3 clade in all bootstrap replicate linear regressions after Bonferroni correction.
(PDF)

**S3 Fig. Inferred inverse rate of coalescence per lineage (effective population size) through time.**
(PDF)

**S4 Fig. For VP1 dataset samples with 'age_range1' data, for all years, the proportion of samples in each age category for each clade.** The over-representation of adults and the elderly in the A2 subclade can be seen clearly, along with the over-representation of adults (and to a lesser extent, the elderly) in the A1 subclade.
(PDF)

**S5 Fig. Mean number of AA mutations in the BC- and DE-loops over time, colored by clade.** In the BC loop, AA positions 90, 92, 95, 97, 98, and 103, were used, and in the DE loop, AA positions 140–146 and 149 were used. The BC-loop plot (top) shows that the B3 clade had around 4 mutations between 2000 and 2008, then about 7 years without mutations. In contrast, the A2 clade had only one mutation prior to 2010, but then had two between 2011 and 2015. Mean mutation count lines have been plotted slightly above and below their true value so that all lines can be seen when they share the same value. Calculated for the whole genome dataset.
(PDF)

**S6 Fig. Most common C-terminus epitope patterns on the VP1 tree.** This figure extends from Fig 4D and 4E, but shows the most common C-terminus patterns at variable positions 280, 283, 284, 288, 290, 297, 299, 301, 304–306, and 308. Cumulative counts of the AA changes from the most recent reliably identifiable sequence (marked with '0') are shown along the branches. As in the BC-loop in Fig 4, the A2 subclade shows substantial recent evolution in this region.
(PDF)

**S7 Fig. Distribution of VP1 sequences by region over time.** The number of VP1 sequences per month from 2010 until the end of 2018 is shown per region. The biennial autumn outbreak pattern in Europe and North America is apparent. The lack of sequences in many regions makes patterns hard to discern.
(PDF)

**S8 Fig. Most common BC- and DE-loop epitope patterns on the VP1 tree.** This figure extends from Fig 4D and 4E, but additionally includes the cumulative counts of amino-acid changes relative to the most recently reliable identifiable sequence (marked with '0') along the branches.
(PDF)

**S9 Fig. Distribution of synonymous and non-synonymous changes in VP1. A)** The distribution of synonymous and non-synonymous changes in VP1 shows that most sites have between 5–10 synonymous changes, but 0 or 1 non-synonymous change, suggesting most of protein is under purifying selection. However, some sites have over 15 non-synonymous changes, suggesting selection for diversification. **B)** When sites are ordered by their number of

non-synonymous changes, codons within the BC- and DE-loops are among the most variable positions.
(PDF)

**S1 Table. Metadata for samples sequenced in this study.**
(TSV)

**S2 Table. AA Mutations in BC- and DE-Loops, C-Terminus, and VP2.** This table extends from Fig 4B, but includes the pattern differences for the C-terminus and VP2 AA positions 135–156.
(XLSX)

**S3 Table. Metadata for all publicly available EV-D68 VP1 sequences included in this study.**
(TSV)

**S4 Table. Metadata for all publicly available EV-D68 full-genome sequences included in this study.**
(TSV)

**S5 Table. Clade definitions for VP1 dataset.**
(XLSX)

**S6 Table. Clade definitions for whole genome dataset.**
(XLSX)

**S7 Table. Regional counts of VP1 sequences.**
(XLSX)

**S8 Table. Regional counts of whole genome sequences.**
(XLSX)

## Acknowledgments

We gratefully acknowledge expert sequencing service and advice by Tanja Normark, Cecilia Svensson, Anna Lyander, Emilia Ottosson Laakso and Valtteri Wirta at the Clinical Genomics Unit at Science for Life Laboratory (SciLifeLab), and expert technical assistance by Lina Thebo at the Karolinska Institute. The team at the University Hospital Basel thanks for excellent technical assistance of Daniela Lang. We would like to thank Moira Zuber for her work on implementing subclade specific mapping and Adam Mazur for his assistance in using PyMOL to generate the crystal structure in Fig 4.

We are very grateful to researchers who shared their previously unpublished age data with us: Amary Falls, Institut Pasteur de Dakar, Senegal; Sindy Böttcher & Sabine Diedrich, Robert Koch Institute, Berlin, Germany; Katzumi Mizuta, Yamagata Prefectural Institute of Public Health, Yamagata, Japan; Sofie Midgley, Statens Serum Institut, Copenhagen, Denmark.

Analyses were run using sciCORE (scicore.unibas.ch) scientific computing core facility at University of Basel.

Finally, we are grateful to scientists and public health authorities globally for sharing EV-D68 sequences and metadata openly and without delay.

## Author Contributions

**Conceptualization:** Emma B. Hodcroft, Robert Dyrdak, Richard A. Neher, Jan Albert.

**Data curation:** Emma B. Hodcroft, Robert Dyrdak, Cristina Andrés, Adrian Egli, Josiane Reist, Diego García Martínez de Artola, Julia Alcoba-Flórez, Hubert G. M. Niesters, Andrés Antón, Randy Poelman, Marijke Reynders, Elke Wollants, Richard A. Neher.

**Formal analysis:** Emma B. Hodcroft, Robert Dyrdak, Richard A. Neher, Jan Albert.

**Investigation:** Richard A. Neher, Jan Albert.

**Methodology:** Emma B. Hodcroft, Robert Dyrdak, Josiane Reist, Diego García Martínez de Artola, Hubert G. M. Niesters, Andrés Antón, Richard A. Neher, Jan Albert.

**Project administration:** Richard A. Neher, Jan Albert.

**Resources:** Cristina Andrés, Adrian Egli, Julia Alcoba-Flórez, Randy Poelman, Marijke Reynders, Elke Wollants.

**Software:** Emma B. Hodcroft, Richard A. Neher.

**Supervision:** Richard A. Neher, Jan Albert.

**Visualization:** Emma B. Hodcroft.

**Writing – original draft:** Emma B. Hodcroft, Robert Dyrdak, Richard A. Neher, Jan Albert.

**Writing – review & editing:** Emma B. Hodcroft, Robert Dyrdak, Cristina Andrés, Adrian Egli, Josiane Reist, Diego García Martínez de Artola, Julia Alcoba-Flórez, Hubert G. M. Niesters, Andrés Antón, Randy Poelman, Marijke Reynders, Elke Wollants, Richard A. Neher, Jan Albert.

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
