## [Decision Letter · Decision Letter 0]

12 Dec 2021

Dear Dr. Hodcroft,

Thank you very much for submitting your manuscript "Evolution, geographic spreading, and demographic distribution of Enterovirus D68" for consideration at PLOS Pathogens. As with all papers reviewed by the journal, your manuscript was reviewed by members of the editorial board and by several independent reviewers. The reviewers appreciated the attention to an important topic. Based on the reviews, we are likely to accept this manuscript for publication, providing that you modify the manuscript according to the review recommendations.

Sincerely,

Nels C. Elde, Ph.D.

Associate Editor

PLOS Pathogens

Adam Lauring

Section Editor

PLOS Pathogens

Kasturi Haldar

Editor-in-Chief

PLOS Pathogens

orcid.org/0000-0001-5065-158X

Michael Malim

Editor-in-Chief

PLOS Pathogens

orcid.org/0000-0002-7699-2064

Reviewer Comments (if any, and for reference):

Reviewer's Responses to Questions

**Part I - Summary**

Reviewer #1: The authors analyse the worldwide epidemiology of enterovirus D68 by analysing 1,987 partial genomes and 718 whole genomes, 53 of which they contribute. First they analyse the genetic structure of the virus populations circulating in 2014, 2016, and 2018 with a focus on the latter, for which they have more metadata. They detect 5 major clades and show that for a given epidemic season, diversification seems to begin 1 to two years before suggesting some undocumented circulation. They also show that the substitutions are not neutral and correlated with amino acid mutations potentially affecting epitope recogbition by antibodies. Furthermore, by factoring in the age of the patients in the analysis they show that all lineages infect young children but that one of the lineages (A2) is more often found in older patients. Finally, they discuss the implications of this study in the wider context of multi-year D68 circulation.

Reviewer #2: Fundamentally, this is a sequencing paper. The Aus had access to an excellent collection of 2018 outbreak isolates, sequenced these as full genomes, then combined/compared the data with a deep set of publicly available partial & complete sequences (mostly VP1), to draw a more accurate picture of circulating clades, especially with regard to old/new patterns in the putative capsid neutralizing epitope regions of VP1 and the age distribution of individuals from whom isolates/clades were circulating.

EV-D68 is current frequent study focus because of the frequency of modern outbreaks, and because an understanding of these outbreak parameters from the virus evolution point of view could help model and predict the behaviors of other emerging pathogens, such as potential novel coronaviruses or other pandemic (as opposed to outbreak) threats. The general significance of this manuscript is that evolutionary predictive potential, based on a broad dataset of real-time outbreak sequences. Strain origins can be tracked from common geographic regions and attributed to specific outbreaks.

The methods used here to collect and sequence samples are routine (these days) and properly applied. The alignments and statistical comparisons are valid, supporting geographic strain tracing and assignment of putative outbreak origins. As a whole, these are useful data additions about the behavior of D-68, and perhaps more generalized to EVs as a genus. Nothing proposed here contradicts the parallel decades of data showing similar tracking and evolutionary trends for (say) the related polio or coxsackie viruses.

Reviewer #3: The paper by Hodcroft and colleagues describes an exceptionally thorough evolutionary analysis of EV-D68 sequences across the globe.  It sheds lights on the factors driving diversification and “hidden” diversification, and suggest that re-infections of adults is a major driver of antigenic evolution. The paper is very well written, with clear figures and very balanced explanations, and I laud the authors for making the data (as always) publicly available and accessible through NextStrain.

**Part II – Major Issues: Key Experiments Required for Acceptance**

Reviewer #1: My knowledge of D68 enterovirus biology and epidemiology is close to zero so I may be missing essential aspects regarding the impact of the work. I found the study was conducted in an ideal way based on the data collected by the authors. I also think this compilation of existing data, especially the effort to identify patient age, can be helpful to others. Finally, the authors' effort to share their analysis with a wide audience is particularly impressive.

My main concern about the study has to do with the uneven nature of the sampling. I also have additional minor concerns.

1) Epidemiological context

I am perhaps not the most appropriate person to review this manuscript because I know little about the biology and epidemiology of D68. But, on the other side, I also perhaps reflect the reaction of a more general audience.

1.1 In the introduction, I would have liked to read some more details about D68, e.g. is it an RNA virus, what is the typical course of infection, what are its transmission routes?

1.2 More on the epidemiological side, it would be interesting to have an idea of variations in incidence on a geographical and temporal scale (e.g. by showing weekly or monthly reports per region of the world). It would also be good to have an idea of the yearly incidence (or of the unknowns regarding this incidence).

1.3 I was wondering whether other studies have used sequence datasets of comparable size to study the worldwide genetic structure and epidemiology of the virus? Because currently, it is difficult to assess the originality of the study with that respect. Furthermore, I was wondering why the analyses focus in particular on 2014, 2016, and 2018 when there seem to be several samples collected before that date.

2) Sampling

As in any epidemiological study, a key issue is sampling. Overall, I think the authors could perhaps improve the demonstration that their sampling is exhaustive.

2.1 The meta-data associated with the 53 original genomes generated in the study is very detailed but for the other genomes and the VP1 sequences the information is limited. Adding a summary table for all of the data would be great to rapidly visualise the coverage per region of the world and per year.

2.2 The temporal coverage seems very variable. For instance, based on Figure 2C, there were 2.5 more sequences collected in 2014 than in 2016 or 2018. Is this related to differences in the magnitude of the yearly epidemics? Could this also explain the lower diversity, as, for instance, in Figure 3B?

2.3 In several occurrences, the authors claim that the sampling is very dense in a geographical region but I think that the balance of this worldwide coverage could be better shown using the data.

2.4 The authors favour a (nice) hypothesis revolving around age-based sensitivity to virus lineages but could the uneven age distribution between the sampling years (2014 vs. 2016 and 2018) be caused by a sampling bias? (see also comment 3.4 below)

3) Evolutionary dynamics

3.1 I was unsure I grasped the message about the (very nice) plot in Figure 2C between lineage coalescence and sampling. It seems intuitive that a peak in sampling would generate a peak in coalescence right before. However, perhaps the relative width of the coalescence peak with respect to the sampling peak could be related to the genetic diversity in the samples?

3.2 My expertise in epitope evolution is very limited but naively I would think that the clustering should be very strong if the authors show amino acid substitutions on a phylogeny. Wouldn't it be possible to have a more phenotypic measure as a label for the tree? That being said, apparently, the DTAQTF seems to revert in the BC-loop, which could be interesting but it is not discussed.

3.3 Still about the epitope evolution, I think some statistical support would be good for the statements regarding the fact that variability is mostly found on the outer surface of the protein.

3.4 As an evolutionary biologist interested in infection virulence, I was very enthusiastic about the correlation found between virus lineage and patient age. However, I have two comments. First, from a statistical point of view, I was surprised to see the authors use a linear model to study the association because it de facto assumes that the lineages are independent. In other words, this is a clear instance where correcting for phylogenetic non-independence is important. Second, Figure 3B suggests that there is an uneven sampling in terms of age between the years. If the authors could show that this association with age is independent of the sampling year, either by adding it as a cofactor in the model or by only using the 2014 year, I think that would strengthen their result.

4) Perspectives

Overall, I found that the strength of the discussion could be improved in at least three ways.

4.1 The authors lean on a preprint from wastewater samples. In addition to the legitimate question about the validity of this work, the method used is itself very experimental. I could be wrong but I expect wastewater samples to be very fragmented and I am unsure about the confidence we can have regarding the genetic composition of the sequences retrieved.

4.2 The paragraph about serology reads also strangely. The authors describe in detail an earlier study by Harrison et al (2019, Emerg Infect Dis) to discuss sera from individuals react differently against viruses based on their ages to support the idea that individuals are reinfected. The problem is that two paragraphs are a bit short to prove this idea. Furthermore, the authors invoke the original antigenic sin (which would benefit from a reference) to explain that older individuals would be infected by lineage A2, but I do not understand why if so we shouldn't see infections in older individuals by other lineages (especially B3 that is more divergent). If this is the motivation of the study, I think it should be present from the introduction and the main hurdles should be explained clearly (i.e. the fact that "young" adults are asymptomatic and rarely sampled).

4.3 I found the mentioning of SARS-CoV-2 strange, partly because there was no scientific reference to back up the claims made, and also partly because of the lack of biological justification. If the authors wish to draw a parallel between D68 and SARS-CoV-2 I think they should make a better case as to how the two are related. Intuitively, I would have said that a reference to seasonal coronaviruses would have been more appropriate here.

Reviewer #2: (No Response)

Reviewer #3: None

**Part III – Minor Issues: Editorial and Data Presentation Modifications**

Reviewer #1: abstract: I am unsure about the formulation of the first sentence. Can we really say that an outbreak of enterovirus cased a disease? Intuitively, I would say that a virus caused an outbreak of disease or that a virus outbreak caused an increase in diseases.

l.47: Add a reference for the 2014 outbreaks?

Figure 1: How were the "key" clusters defined?

l.130: Please further explain why the epidemic was "well-sampled" in 2015-2016.

l.132-133: I was unable to determine whether this sentence was refering to one of the figures.

l.133-134: Same comment here: can this be back-up by one of the figures?

l.137-138: Does this correspond to Figure 1B?

l.141-148: Knowing that the lineage originates in early 2017, it is probably feasible to estimate how much secondary infections this initial "common ancestor" would have generated early 2018. I think it would help to determine whether it would be feasible to have this lineage represented (here the authors do have 3 full genomes from early 2018).

l.169: Here it would be interesting to know more about the virus circulation. Is it like influenza where Asia acts as a reservoir? Is the virus seasonal in Europe and the USA? In this case, "underreporting" in a country could also be due to the fact that there are no cases in this country.

l.195: It would be good to provide quantitative support for the claim that the continents were "thoroughly sampled".

l.300: Titers refers to antibody titers correct?

l.305: A high substitution rate compared to what?

l.345: I think a reference is needed for the original antigenic sin

Some of the figures were difficult to interpret. For instance, in Figure 2B and 2C it is unclear what lineages are considered (are these all the lineages shown in Figure 1A?). Seapking of Figure 1A, it would actually help to label the panels so that they can be mentionned in the text.

l.645: What was the substitution model used for the phylogeny inference?

Suppl Table IV: What does ?10^3 mean in the coverage column?

[signed and dated]

- Samuel Alizon, 19 Nov 2021

Reviewer #2: Points the Aus may wish to consider.

A key conclusion here is that D68 is apparently under “continuous antigenic evolution”, a not unexpected finding or a novel one. To “predict which pathogen may lead to the next viral pandemic” requires a very different set of judgements that are not answered here. Pandemics require one (or more) novel, transmissible serotypes to be introduced into an immunologically naive population.

1. EV-D68 is a defined EV genotype, meaning isolates are binned here by sequencing similarity thresholds not (as in old days), serotyping. Certainly sequence clades A, B, C are under positive selection as measured by diversity in certain loop elements, but is there any evidence that fresh convalescent human serum from any recently infected individual (e.g. clade A), will not cross-neutralize viruses from the other clades? Doesnt have to be the same high titer, but will it cross-neutralize at even a partially protective titer (perhaps 1:5)?

2. If “no” then all presented data really just support the idea that D68 as a serotype is not evolving linearly, but may just continually recycle some immunogenic, permissible loop configurations, albeit many decades apart. Flu does this, for example, and leaves the same type of host age-dependent patterns as the clades rise, fall, and eventually recycle. By definition, this evolution pattern may lead to continuous outbreaks in temporarily naive geographic regions, but not pandemics

3. If “yes” then are the data telling us that D-68 is really on the verge of becoming “D-69”, or “D-98” or something else that is so immunologically distinct and different, it has the potential for a novel pandemic?

4. Are there known structural constraints on the capsid or receptor binding that might differentiate between “yes” and “no”?

5. I might have missed it, but do the Nim2 and Nim3 epitope sequences (check the sequence analogs in polio or the RVs) show similar patterns of clade-specific variability?

Reviewer #3: 1.     In the introduction, the authors write: “ .. the degree to which EV-D68 evolution is driven by immune escape or whether incidence patterns are determined by pre-existing immunity is unclear”. I could not see much distinction between these two, perhaps rephrase.

2.     The analysis of antigenic evolution (pages 11 onward) is very descriptive and less quantitative. For example, I would have expected a site-specific dn/ds analysis here that can quantify positive selection or even contrast between rate of evolution of different sites. Also, it would have been nice to see a quantitative measure of the imbalance of the phylogeny, which once again supports the notion of antigenic drift.

3.     Would be nice to add to panels D and E of figure 4, how many amino-acid replacements are there between the different color coded epitope?

4.     The authors discuss how antigenic evolution drives diversification of the virus, and also mention the biennial  pattern of infection of this virus. Antigenic evolution has been described for many viruses (influenza, seasonal coronaviruses). Could the authors discuss and raise hypotheses why there is a biennial pattern (rather than annual) pattern and if and how this is related to re-infections of adults?

PLOS authors have the option to publish the peer review history of their article (what does this mean?). If published, this will include your full peer review and any attached files.

Reviewer #1: **Yes: **Samuel Alizon

Reviewer #2: No

Reviewer #3: No

Figure Files:

Data Requirements:

Reproducibility:

References:

---

## [Editor Report · Decision Letter 1]

10 Apr 2022

Dear Dr. Hodcroft,

We are pleased to inform you that your manuscript 'Evolution, geographic spreading, and demographic distribution of Enterovirus D68' has been provisionally accepted for publication in PLOS Pathogens.

Best regards,

Nels C. Elde, Ph.D.

Associate Editor

PLOS Pathogens

Adam Lauring

Section Editor

PLOS Pathogens

Kasturi Haldar

Editor-in-Chief

PLOS Pathogens

orcid.org/0000-0001-5065-158X

Michael Malim

Editor-in-Chief

PLOS Pathogens

orcid.org/0000-0002-7699-2064
---

## [Editor Report · Acceptance letter]

19 May 2022

Dear Dr. Hodcroft,

We are delighted to inform you that your manuscript, "Evolution, geographic spreading, and demographic distribution of Enterovirus D68," has been formally accepted for publication in PLOS Pathogens.

Best regards,

Kasturi Haldar

Editor-in-Chief

PLOS Pathogens

orcid.org/0000-0001-5065-158X

Michael Malim

Editor-in-Chief

PLOS Pathogens

orcid.org/0000-0002-7699-2064